# Beyond Pixel Norm-Balls: Parametric Adversaries using an Analytically Differentiable Renderer

**Hsueh-Ti Derek Liu**
University of Toronto
hsuehtil@cs.toronto.edu

**Michael Tao**
University of Toronto
mtao@dgp.toronto.edu

**Chun-Liang Li**
Carnegie Mellon University
chunlial@cs.cmu.edu

**Derek Nowrouzezahrai**
McGill University
derek@cim.mcgill.ca

**Alec Jacobson**
University of Toronto
jacobson@cs.toronto.edu

## Abstract

Many machine learning image classifiers are vulnerable to adversarial attacks, inputs with perturbations designed to intentionally trigger misclassification. Current adversarial methods directly alter pixel colors and evaluate against *pixel norm-balls*: pixel perturbations smaller than a specified magnitude, according to a measurement norm. This evaluation, however, has limited practical utility since perturbations in the pixel space do not correspond to underlying real-world phenomena of image formation that lead to them and has no security motivation attached. Pixels in natural images are measurements of light that has interacted with the geometry of a physical scene. As such, we propose a novel evaluation measure, *parametric norm-balls*, by directly perturbing physical parameters that underlie image formation. One enabling contribution we present is a physically-based differentiable renderer that allows us to propagate pixel gradients to the parametric space of lighting and geometry. Our approach enables physically-based adversarial attacks, and our differentiable renderer leverages models from the interactive rendering literature to balance the performance and accuracy trade-offs necessary for a memory-efficient and scalable adversarial data augmentation workflow.

## 1 Introduction

Research in adversarial examples continues to contribute to the development of robust (semi-)supervised learning (Miyato et al., 2018), data augmentation (Goodfellow et al., 2015; Sun et al., 2018), and machine learning understanding (Kanbak et al., 2018). One important caveat of the approach pursued by much of the literature in adversarial machine learning, as discussed recently (Goodfellow,

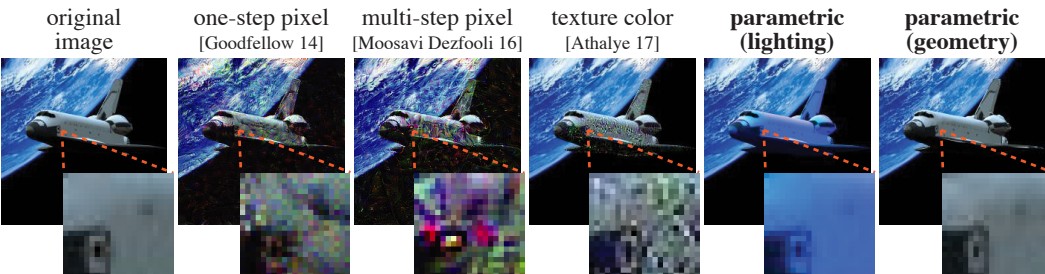

Figure 1: Traditional pixel-based adversarial attacks yield unrealistic images under a larger perturbation ($L^\infty$-norm $\approx 0.82$), however our parametric lighting and geometry perturbations output more realistic images under the same norm (more results in Appendix A).

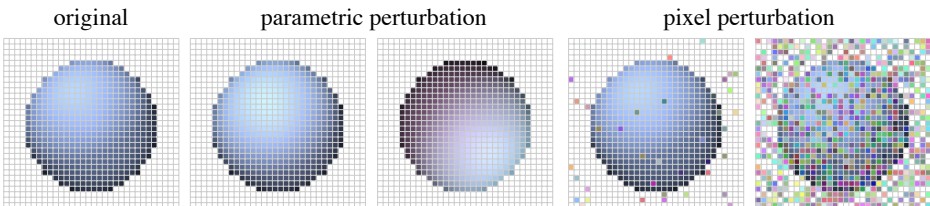

Figure 2: Parametrically-perturbed images remain natural, whereas pixel-perturbed ones do not.

2018; Gilmer et al., 2018), is the reliance on overly simplified attack metrics: namely, the use of pixel value differences between an adversary and an input image, also referred to as the pixel *norm-balls*.

The pixel norm-balls game considers pixel perturbations of norm-constrained magnitude (Goodfellow et al., 2015), and is used to develop adversarial attackers, defenders and training strategies. The pixel norm-ball game is attractive from a research perspective due to its simplicity and well-posedness: no knowledge of image formation is required and any arbitrary pixel perturbation remains eligible (so long as it is "small", in the perceptual sense). Although the pixel norm-ball is useful for research purposes, it only captures limited real-world security scenarios.

Despite the ability to devise effective adversarial methods through the direct employment of optimizations using the pixel norm-balls measure, the pixel manipulations they promote are divorced from the types of variations present in the real world, limiting their usefulness "in the wild". Moreover, this methodology leads to defenders that are only effective when defending against unrealistic images/attacks, not generalizing outside of the space constrained by pixel norm-balls. In order to consider conditions that enable adversarial attacks in the real world, we advocate for a new measurement norm that is rooted in the physical processes that underly realistic image synthesis, moving away from overly simplified metrics, e.g., pixel norm-balls.

Our proposed solution – *parametric norm-balls* – rely on perturbations of physical parameters of a synthetic image formation model, instead of pixel color perturbations (Figure 2). To achieve this, we use a physically-based differentiable renderer which allows us to perturb the underlying parameters of the image formation process. Since these parameters *indirectly* control pixel colors, perturbations in this parametric space *implicitly* span the space of natural images. We will demonstrate two advantages that fall from considering perturbations in this parametric space: (1) they enable adversarial approaches that more readily apply to real-world applications, and (2) they permit the use of much more significant perturbations (compared to pixel norms), without invalidating the realism of the resulting image (Figure 1). We validate that parametric norm-balls game playing is critical for a variety of important adversarial tasks, such as building defenders robust to perturbations that can occur naturally in the real world.

We perform perturbations in the underlying image formation parameter space using a novel physically-based differentiable renderer. Our renderer analytically computes the derivatives of pixel color with respect to these physical parameters, allowing us to extend traditional pixel norm-balls to physically-valid parametric norm-balls. Notably, we demonstrate perturbations on an environment's *lighting* and on the shape of the 3D *geometry* it shades. Our differentiable renderer achieves state-of-the-art performance in speed and scalability (Section 3) and is fast enough for *rendered adversarial data augmentation* (Section 5): training augmented with adversarial images generated with a renderer.

Existing differentiable renders are slow and do not scalable to the volume of high-quality, high-resolutions images needed to make adversarial data augmentation tractable (Section 2). Given our analytically-differentiable renderer (Section 3), we are able to demonstrate the efficacy of parametric space perturbations for generating adversarial examples. These adversaries are based on a substantially different phenomenology than their pixel norm-balls counterparts (Section 4). Ours is among the first steps towards the deployment of rendered adversarial data augmentation in real-world applications: we train a classifier with computer-generated adversarial images, evaluating the performance of the training against real photographs (i.e., captured using cameras; Section 5). We test on real photos to show the parametric adversarial data augmentation increases the classifier's robustness to "deformations" happened in the real world. Our evaluation differs from the majority of existing literature which evaluates against computer-generated adversarial images, since our parametric space perturbation is no-longer a wholly idealized representation of the image formation model but, instead, modeled against of theory of realistic image generation.

## 2    RELATED WORK

Our work is built upon the fact that *simulated* or *rendered* images can participate in computer vision and machine learning on real-world tasks. Many previous works use rendered (simulated) data to train deep networks, and those networks can be deployed to real-world or even outperform the state-of-the-art networks trained on real photos (Movshovitz-Attias et al., 2016; Chen et al., 2016; Varol et al., 2017; Su et al., 2015; Johnson-Roberson et al., 2017; Veeravasarapu et al., 2017b; Sadeghi & Levine, 2016; James & Johns, 2016). For instance, Veeravasarapu et al. (2017a) show that training with 10% real-world data and 90% simulation data can reach the level of training with full real data. Tremblay et al. (2018) even demonstrate that the network trained on synthetic data yields a better performance than using real data alone. As rendering can cheaply provide a theoretically infinite supply of annotated input data, it can generate data which is orders of magnitude larger than existing datasets. This emerging trend of training on synthetic data provides an exciting direction for future machine learning development. Our work complements these works. We demonstrate the utility of rendering can be used to study the potential danger lurking in misclassification due to subtle changes to geometry and lighting. This provides a future direction of combining with synthetic data generation pipelines to perform *physically based adversarial training on synthetic data*.

**Adversarial Examples**    Szegedy et al. (2014) expose the vulnerability of modern deep neural nets using purposefully-manipulated images with human-imperceptible misclassification-inducing noise. Goodfellow et al. (2015) introduce a fast method to harness adversarial examples, leading to the idea of pixel norm-balls for evaluating adversarial attackers/defenders. Since then, many significant developments in adversarial techniques have been proposed (Akhtar & Mian, 2018; Szegedy et al., 2014; Rozsa et al., 2016; Kurakin et al., 2017; Moosavi Dezfooli et al., 2016; Dong et al., 2018; Papernot et al., 2017; Moosavi-Dezfooli et al., 2017; Chen et al., 2017; Su et al., 2017). Our work extends this progression in constructing adversarial examples, a problem that lies at the foundation of adversarial machine learning. Kurakin et al. (2016) study the transferability of attacks to the physical world by printing then photographing adversarial *images*. Athalye et al. (2017) and Eykholt et al. (2018) propose extensions to non-planar (yet, still fixed) geometry and multiple viewing angles. These works still rely fundamentally on the direct pixel or texture manipulation on physical objects. Since these methods assume independence between pixels in the image or texture space they remain variants of pixel norm-balls. This leads to unrealistic attack images that cannot model real-world scenarios (Goodfellow, 2018; Hendrycks & Dietterich, 2018; Gilmer et al., 2018). Zeng et al. (2017) generate adversarial examples by altering physical parameters using a rendering network (Liu et al., 2017) trained to approximate the physics of realistic image formation. This data-driven approach leads to an image formation model biased towards the rendering style present in the training data. This method also relies on differentiation through the rendering network in order to compute adversaries, which requires high-quality training on a large amount of data. Even with perfect training, in their reported performance, it still requires 12 minutes on average to find new adversaries, we only take a few seconds Section 4.1. Our approach is based on a differentiable physically-based renderer that directly (and, so, more convincingly) models the image formation process, allowing us to alter physical parameters – like geometry and lighting – and compute derivatives (and adversarial examples) much more rapidly compared to the (Zeng et al., 2017). We summarize the difference between our approach and the previous non-image adversarial attacks in Table 1.

Table 1: Previous non-pixel attacks fall short in either the parameter range they can take derivatives or the performance.

| Methods | Perf. | Color | Normal | Material | **Light** | **Geo.** |
|---|---|---|---|---|---|---|
| Athalye 17 | ✓ | ✓ | | | | |
| Zeng 17 | | | ✓ | ✓ | ✓ | |
| Ours | ✓ | ✓ | ✓ | | ✓ | ✓ |

**Differentiable Renderer**    Applying parametric norm-balls requires that we differentiate the image formation model with respect to the physical parameters of the image formation model. Modern realistic computer graphics models do not expose facilities to directly accommodate the computation of derivatives or automatic differentiation of pixel colors with respect to geometry and lighting variables. A physically-based

Table 2: Previous differentiable renderers fall short in one way or another among Performance, Bias, or Accuracy.

| Methods | Perf. | Unbias | Accu. |
|---|---|---|---|
| NN proxy  (Liu 17) | ✓ | | |
| Approx.    (Kato 18) | ✓ | ✓ | |
| Autodiff    (Loper 14) | | ✓ | ✓ |
| Analytical (Ours) | ✓ | ✓ | ✓ |

differentiable renderer is fundamental to computing derivative of pixel colors with respect to scene parameters and can benefit machine learning in several ways, including promoting the development of novel network architectures (Liu et al., 2017), in computing adversarial examples (Athalye et al., 2017; Zeng et al., 2017), and in generalizing neural style transfer to a 3D context (Kato et al., 2018; Liu et al., 2018). Recently, various techniques have been proposed to obtain these derivatives: Wu et al. (2017); Liu et al. (2017); Eslami et al. (2016) use neural networks to *learn* the image formation process provided a large amount of input/output pairs. This introduces unnecessary bias in favor of the training data distribution, leading to inaccurate derivatives due to imperfect learning. Kato et al. (2018) propose a differentiable renderer based on a simplified image formation model and an underlying linear approximation. Their approach requires no training and is unbiased, but their approximation of the image formation and the derivatives introduce more errors. Loper & Black (2014); Genova et al. (2018) use automatic differentiation to build fully differentiable renderers. These renderers, however, are expensive to evaluate, requiring orders of magnitude more computation and much larger memory footprints compared to our method.

Our novel differentiable renderer overcomes these limitations by efficiently computing *analytical* derivatives of a physically-based image formation model. The key idea is that the non-differentiable visibility change can be ignored when considering infinitesimal perturbations. We model image variations by changing geometry and realistic lighting conditions in an analytically differentiable manner, relying on an accurate model of diffuse image formation that extend spherical harmonics-based shading methods (Appendix C). Our analytic derivatives are efficient to evaluate, have scalable memory consumption, are unbiased, and are accurate by construction (Table 2). Our renderer explicitly models the physics of the image formation processes, and so the images it generates are realistic enough to illicit correct classifications from networks trained on real-world photographs.

## 3 ADVERSARIAL ATTACKS IN PARAMETRIC SPACES

Adversarial attacks based on pixel norm-balls typically generate adversarial examples by defining a cost function over the space of images $\mathcal{C} : I \rightarrow \mathbb{R}$ that enforces some intuition of what failure should look like, typically using variants of gradient descent where the gradient $\partial \mathcal{C}/\partial I$ is accessible by differentiating through networks (Szegedy et al., 2014; Goodfellow et al., 2015; Rozsa et al., 2016; Kurakin et al., 2017; Moosavi Dezfooli et al., 2016; Dong et al., 2018).

The choices for $\mathcal{C}$ include increasing the cross-entropy loss of the correct class (Goodfellow et al., 2015), decreasing the cross-entropy loss of the least-likely class (Kurakin et al., 2017), using a combination of cross-entropies (Moosavi Dezfooli et al., 2016), and more (Szegedy et al., 2014; Rozsa et al., 2016; Dong et al., 2018; Tramèr et al., 2017). We combine of cross-entropies to provide flexibility for choosing untargeted and targeted attacks by specifying a different set of labels:

$$\mathcal{C}\big(I(U,V)\big) = -\text{CrossEntropy}\big(f(I(U,V)), L_d\big) + \text{CrossEntropy}\big(f(I(U,V)), L_i\big), \quad (1)$$

where $I$ is the image, $f(I)$ is the output of the classifier, $L_d, L_i$ are labels which a user wants to *decrease* and *increase* the predicted confidences respectively. In our experiments, $L_d$ is the correct class and $L_i$ is either ignored or chosen according to user preference. Our adversarial attacks in the parametric space consider an image $I(U,V)$ is the function of physical parameters of the image formation model, including the lighting $U$ and the geometry $V$. Adversarial examples constructed by perturbing physical parameters can then be computed via the chain rule

$$\frac{\partial \mathcal{C}}{\partial U} = \frac{\partial \mathcal{C}}{\partial I}\frac{\partial I}{\partial U} \qquad\qquad \frac{\partial \mathcal{C}}{\partial V} = \frac{\partial \mathcal{C}}{\partial I}\frac{\partial U}{\partial V}, \quad (2)$$

where $\partial I/\partial U, \partial I/\partial V$ are derivatives with respect to the physical parameters and we evaluate using our physically based differentiable renderer. In our experiments, we use gradient descent for finding parametric adversarial examples where the gradient is the direction of $\partial I/\partial U, \partial I/\partial V$.

### 3.1 PHYSICALLY BASED DIFFERENTIABLE RENDERER

Rendering is the process of generating a 2D image from a 3D scene by simulating the physics of light. Light sources in the scene emit photons that then interact with objects in the scene. At each interaction, photons are either reflected, transmitted or absorbed, changing trajectory and repeating until arriving at a sensor such as a camera. A physically based renderer models the interactions mathematically (Pharr et al., 2016), and our task is to analytically differentiate the physical process.

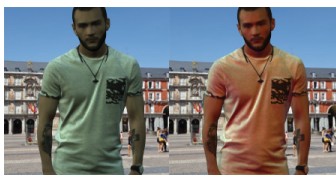

**Top 5:**
miniskirt 28%
t-shirt 21%
boot 6%
crutch 5%
sweatshirt 5%

t-shirt 86%

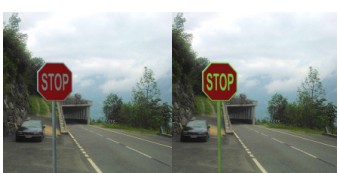

**Top 5:**
water tower 48%
street sign 18%
mailbox 9%
gas pump 3%
barn 3%

street sign 57%

Figure 4: By changing the lighting, we fool the classifier into seeing miniskirt and water tower, demonstrating the existence of adversarial lighting.

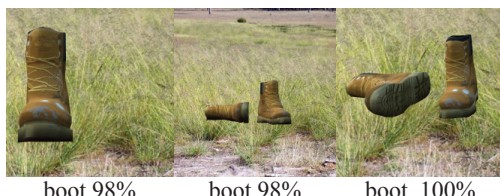

boot 98%          boot 98%          boot 100%

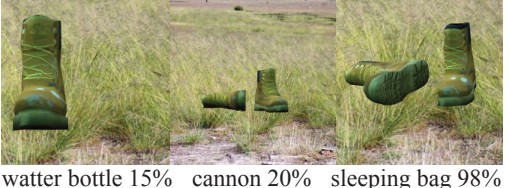

watter bottle 15%    cannon 20%    sleeping bag 98%

Figure 5: We construct a single lighting condition that can simultaneously fool the classifier viewing from different angles.

We develop our differentiable renderer with common assumptions in real-time rendering (Akenine-Moller et al., 2008) – diffuse material, local illumination, and distant light sources. Our diffuse material assumption considers materials which reflect lights uniformly for all directions, equivalent to considering non-specular objects. We assume that variations in the material (texture) are piece-wise constant with respect to our triangle mesh discretization. The local illumination assumption only considers lights that bounce directly from the light source to the camera. Lastly, we assume light sources are far away from the scene, allowing us to represent lighting with one spherical function. For a more detailed rationale of our assumptions, we refer readers to Appendix B).

These assumptions simplify the complicated integral required for rendering (Kajiya, 1986) and allow us to represent lighting in terms of *spherical harmonics*, an orthonormal basis for spherical functions analogous to Fourier transformation. Thus, we can *analytically* differentiate the rendering equation to acquire derivatives with respect to lighting, geometry, and texture (derivations found in Appendix C).

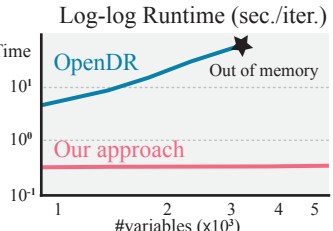

Figure 3: Our differentiable renderer based on analytical derivatives is faster and more scalable than the previous method.

Using analytical derivatives avoids pitfalls of previous differentiable renderers (see Section 2) and make our differentiable renderer orders of magnitude faster than the previous fully differentiable renderer OPENDR (Loper & Black, 2014) (see Figure 3). Our approach is scalable to handle problems with more than 100,000 variables, while OPENDR runs out of memory for problems with more than 3,500 variables.

## 3.2   ADVERSARIAL LIGHTING AND GEOMETRY

*Adversarial lighting* denotes adversarial examples generated by changing the spherical harmonics lighting coefficients $U$ (Green, 2003). As our differentiable renderer allows us to compute $\partial I/\partial U$ analytically (derivation is provided in Appendix C.4), we can simply apply the chain rule:

$$U \leftarrow U - \gamma \frac{\partial \mathcal{C}}{\partial I} \frac{\partial I}{\partial U}, \tag{3}$$

where $\partial \mathcal{C}/\partial I$ is the derivative of the cost function with respect to pixel colors and can be obtained by differentiating through the network. Spherical harmonics act as an implicit constraint to prevent unrealistic lighting because natural lighting environments everyday life are dominated by low-frequency signals. For instance, rendering of diffuse materials can be approximated with only 1% pixel intensity error by the first 2 orders of spherical harmonics (Ramamoorthi & Hanrahan, 2001). As computers can only represent a finite number of coefficients, using spherical harmonics for lighting implicitly filters out high-frequency, unrealistic lightings. Thus, perturbing the parametric space of spherical harmonics lighting gives us more realistic compared to image-pixel perturbations Figure 1.

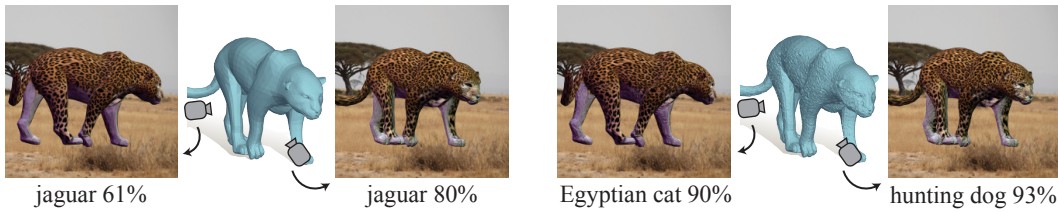

jaguar 61%     jaguar 80%     Egyptian cat 90%     hunting dog 93%

Figure 6: By specifying different target labels, we can create an optical illusion: a jaguar is classified as cat and dog from two different views after geometry perturbations.

*Adversarial geometry* is an adversarial example computed by changes the position of the shape's surface. The shape is encoded as a triangle mesh with $|V|$ vertices and $|F|$ faces, surface points are vertex positions $V \in \mathbb{R}^{|V| \times 3}$ which determine per-face normals $N \in \mathbb{R}^{|F| \times 3}$ which in turn determine the shading of the surface. We can compute adversarial shapes by applying the chain rule:

$$V \leftarrow V - \gamma \frac{\partial \mathcal{C}}{\partial I} \frac{\partial I}{\partial N} \frac{\partial N}{\partial V}, \qquad (4)$$

where $\partial I / \partial N$ is computed via a derivation in Appendix E. Each triangle only has one normal on its face, making $\partial N / \partial V$ computable analytically. In particular, the $3 \times 3$ Jacobian of a unit face normal vector $\mathbf{n}_i \in \mathbb{R}^3$ of the $j$th face of the triangle mesh $V$ with respect to one of its corner vertices $\mathbf{v}_j \in \mathbb{R}^3$ is

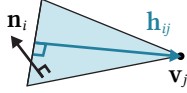

$$\frac{\partial \mathbf{n}_i}{\partial \mathbf{v}_j} = \frac{\mathbf{h}_{ij} \mathbf{n}_i^\mathsf{T}}{\|\mathbf{h}_{ij}\|^2},$$

where $\mathbf{h}_{ij} \in \mathbb{R}^3$ is the height vector: the shortest vector to the corner $\mathbf{v}_j$ from the opposite edge.

## 4 RESULTS AND EVALUATION

We have described how to compute adversarial examples by parametric perturbations, including lighting and geometry. In this section, we show that adversarial examples exist in the parametric spaces, then we analyze the characteristics of those adversaries and parametric norm-balls.

We use $49 \times 3$ spherical harmonics coefficients to represent environment lighting, with an initial real-world lighting condition (Ramamoorthi & Hanrahan, 2001). Camera parameters and the background images are empirically chosen to have correct initial classifications and avoid synonym sets. In Figure 4 we show that single-view adversarial lighting attack can fool the classifier (pre-trained ResNet-101 on ImageNet (He et al., 2016)). Figure 5 shows multi-view adversarial lighting, which optimizes the summation of the cost functions for each view, thus the gradient is computed as the summation over all camera views:

$$U \leftarrow U - \sum_{i \in \text{cameras}} \gamma \frac{\partial \mathcal{C}}{\partial I_i} \frac{\partial I_i}{\partial U}. \qquad (5)$$

If one is interested in a more specific subspace, such as outdoor lighting conditions governed by sunlight and weather, our adversarial lighting can adapt to it. In Figure 7, we compute adversarial lights over the space of skylights by applying one more chain rule to the *Preetham skylight* parameters (Preetham et al., 1999; Habel et al., 2008). Details about taking these derivatives are provided in Appendix D. Although adversarial skylight exists, its low degrees of freedom (only three parameters) makes it more difficult to find adversaries.

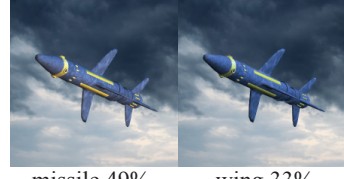

missile 49%     wing 33%

Figure 7: Even if we further constrain to a lighting subspace, skylight, we can still find adversaries.

In Figure 8 and Figure 9 we show the existence of adversarial geometry in both single-view and multi-view cases. Note that we upsample meshes to have >10K vertices as a preprocessing step to increase the degrees of freedom available for perturbations. Multi-view adversarial geometry enables us to perturb the same 3D shape from different viewing directions, which enables us to construct a *deep optical illusion*: The same 3D shape are classified differently from different angles. To create the optical illusion in Figure 6, we only need to specify the $L_i$ in Equation (1) to be a dog and a cat for two different views.

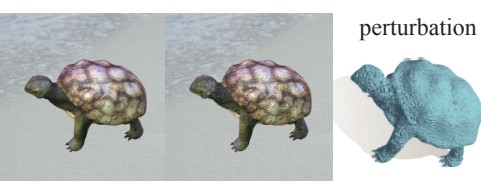 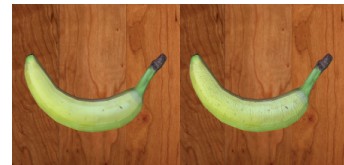 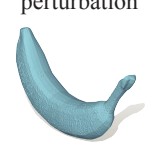

perturbation    perturbation

loggerhead    **Top 3:** assault rifle 87%, military    banana 99%    **Top 3:** slug 91%,
turtle 67%    uniform 6%, six-gun 1%    roundworm 3%, banana 1%

Figure 8: Perturbing points on 3D shapes fools the classifier into seeing rifle/slug.

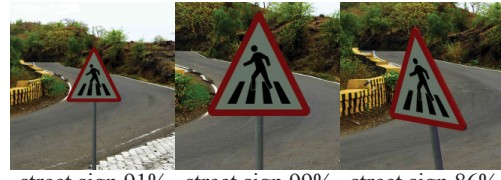
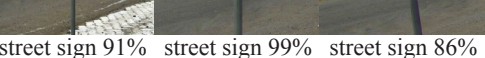
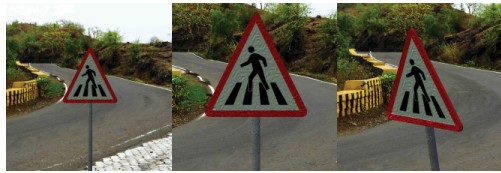

street sign 91%  street sign 99%  street sign 86%     mailbox 51%  mailbox 61%  mailbox 71%

Figure 9: We construct a single adversarial geometry that fools the classifier seeing a mailbox from different angles.

### 4.1 Properties of Parametric Norm-Balls and Adversaries

To further understand parametric adversaries, we analyze how do parametric adversarial examples generalize to black-box models. In Table 3, we test 5,000 ResNet parametric adversaries on unseen networks including AlexNet (Krizhevsky et al., 2012), DenseNet (Huang et al., 2017), SqueezeNet (Iandola et al., 2016), and VGG (Simonyan & Zisserman, 2014). Our result shows that parametric adversarial examples also share across models.

In addition to different models, we evaluate parametric adversaries on black-box viewing directions. This evaluation mimics the real-world scenario that a self-driving car would "see" a stop sign from different angles while driving. In Table 4, we randomly sample 500 correctly classified views for a given shape and perform adversarial lighting and geometry algorithms only on a subset of views, then evaluate the resulting adversarial lights/shapes on all the views. The results show that adversarial lights are more generalizable to fool unseen views; adversarial shapes, yet, are less generalizable.

Switching from pixel norm-balls to parametric norm-balls only requires to change the norm-constraint from the pixel color space to the parametric space. For instance, we can perform a quantitative comparison between parametric adversarial and random perturbations in Figure 10. We use $L^\infty$-$norm = 0.1$ to constraint the perturbed magnitude of each lighting coefficient, and $L^\infty$-$norm = 0.002$ to constrain the maximum displacement of surface points along each axis. The results show how many parametric adversaries can fool the classifier out of 10,000 adversarial lights and shapes respectively. Not only do the

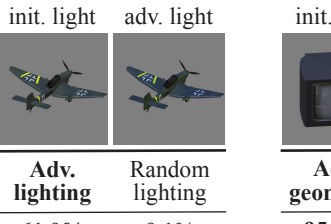

| init. light | adv. light | | init. geo. | adv. geo. |
|---|---|---|---|---|
| **Adv. lighting** | Random lighting | | **Adv. geometry** | Random geometry |
| **61.0%** | 9.1% | | **95.3%** | 17.5% |

Figure 10: A quantitative comparison using parametric norm-balls shows the fact that adversarial lighting/geometry perturbations have a higher success rate (%) in fooling classifiers comparing to random perturbations in the parametric spaces.

parametric norm-balls show the effectiveness of adversarial perturbation, evaluating robustness using parametric norm-balls has real-world implications.

Table 3: We evaluate ResNet adversaries on unseen models and show that parametric adversarial examples also share across models. The table shows the success rate of attacks (%).

|  | Alex | VGG | Squeeze | Dense |
|---|---|---|---|---|
| Lighting | 81.2% | 65.0% | 78.6% | 43.5% |
| Geometry | 70.3% | 58.9% | 71.1% | 40.1% |

Table 4: We compute parametric adversaries using a subset of views (#Views) and evaluate the success rates (%) of attacks on unseen views.

| #Views | 0 | 1 | 5 |
|---|---|---|---|
| Lighting | 0.0% | 29.4% | 64.2% |
| Geometry | 0.0% | 0.6% | 3.6% |

**Runtime** The inset presents our runtime per iteration for computing derivatives. An adversary normally requires less than 10 iterations, thus takes a few seconds. We evaluate our CPU PYTHON implementation and the OPENGL rendering, on an Intel Xeon 3.5GHz CPU with 64GB of RAM and an NVIDIA GeForce GTX 1080. Our runtime depends on the number of pixels requiring derivatives.

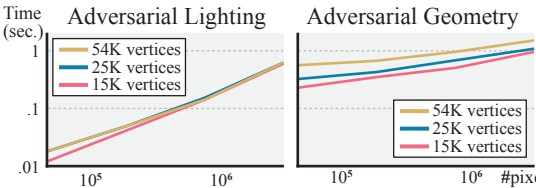

## 5 RENDERED ADVERSARIAL DATA AUGMENTATION AGAINST REAL PHOTOS

We inject adversarial examples, generated using our differentiable renderer, into the training process of modern image classifiers. Our goal is to increase the robustness of these classifiers to real-world perturbations. Traditionally, adversarial training is evaluated against computer-generated adversarial images (Kurakin et al., 2017; Madry et al., 2018; Tramèr et al., 2017). In contrast, our evaluation differs from the majority of the literature, as we evaluate performance against *real photos* (i.e., images captured using a camera), and not computer-generated images. This evaluation method is motivated by our goal of increasing a classifier's robustness to "perturbations" that occur in the real world and result from the physical processes underlying real-world image formation. We present preliminary steps towards this objective, resolving the lack of realism of pixel norm-balls and evaluating our augmented classifiers (i.e., those trained using our rendered adversaries) against real photographs.

**Training** We train the WideResNet (16 layers, 4 wide factor) (Zagoruyko & Komodakis, 2016) on CIFAR-100 (Krizhevsky & Hinton, 2009) augmented with adversarial lighting examples. We apply a common adversarial training method that adds a fixed number of adversarial examples each epoch (Goodfellow et al., 2015; Kurakin et al., 2017). We refer readers to Appendix F for the training detail. In our experiments, we compare three training scenarios: (1) CIFAR-100, (2) CIFAR-100 + 100 images under random lighting, and (3) CIFAR-100 + 100 images under adversarial lighting. Comparing to the accuracy reported in (Zagoruyko & Komodakis, 2016), WideResNets trained on these three cases all have comparable performance ($\approx 77\%$) on the CIFAR-100 test set.

**Testing** We create a test set of real photos, captured in a laboratory setting with controlled lighting and camera parameters: we photographed oranges using a calibrated Prosilica GT 1920 camera under different lighting conditions, each generated by projecting different lighting patterns using an LG PH550 projector. This hardware lighting setup projects lighting patterns from a fixed solid angle of directions onto the scene objects. Figure 11 illustrates samples from the 500 real photographs of our dataset. We evaluate the robustness of our classifier models according to test accuracy. Of note, average prediction accuracies over five trained WideResNets on our test data under the three training cases are (1) 4.6%, (2) 40.4%, and (3) **65.8**%. This result

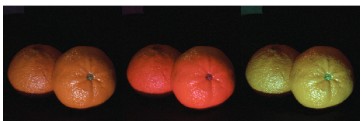

Figure 11: Unlike much of the literature on adversarial training, we evaluate against real photos (captured by a camera), not computer-generated images. This figure illustrates a subset of our test data.

supports the fact that training on rendered images can improve the networks' performance on real photographs. Our preliminary experiments motivate the potential of relying on rendered adversarial training to increase the robustness to visual phenomena present in the real-world inputs.

## 6 LIMITATIONS & FUTURE WORK

Using parametric norm-balls to remove the lack of realism of pixel norm-balls is only the first step to bring adversarial machine learning to real-world. More evaluations beyond the lab experimental data could uncover the potential of the rendered adversarial data augmentation. Coupling the differentiable renderer with methods for reconstructing 3D scenes, such as (Veeravasarapu et al., 2017b; Tremblay et al., 2018), has the potential to develop a complete pipeline for rendered adversarial training. We can take a small set of real images, constructing 3D virtual scenes which have real image statistics, using our approach to manipulate the predicted parameters to construct the parametric adversarial

examples, then perform rendered adversarial training. This direction has the potential to produce limitless simulated adversarial data augmentation for real-world tasks.

Our differentiable renderer models the change of realistic environment lighting and geometry. Incorporating real-time rendering techniques from the graphics community could further improve the quality of rendering. Removing the locally constant texture assumption could improve our results. Extending the derivative computation to materials could enable "adversarial materials". Incorporating derivatives of the visibility change and propagating gradient information to shape skeleton could also create "adversarial poses". These extensions offer a set of tools for modeling real security scenarios. For instance, we can train a self-driving car classifier that can robustly recognize pedestrians under different poses, lightings, and cloth deformations.

### ACKNOWLEDGMENTS

This work is funded in part by NSERC Discovery Grants (RGPIN–2017–05235 & RG-PAS–2017–507938), Connaught Funds (NR2016–17), the Canada Research Chairs Program, the Fields Institute, and gifts by Adobe Systems Inc., Autodesk Inc., MESH Inc. We thank members of Dynamic Graphics Project for feedback and draft reviews; Wenzheng Chen for photography equipments; Colin Raffel and David Duvenaud for discussions and feedback.

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

# Supplementary Material

## A  COMPARISON BETWEEN PERTURBATION SPACES

We extend our comparisons against pixel norm-balls methods (Figure 1) by visualizing the results and the generated perturbations (Figure 12). We hope this figure elucidates that our parametric perturbation are more realistic several scales of perturbations.

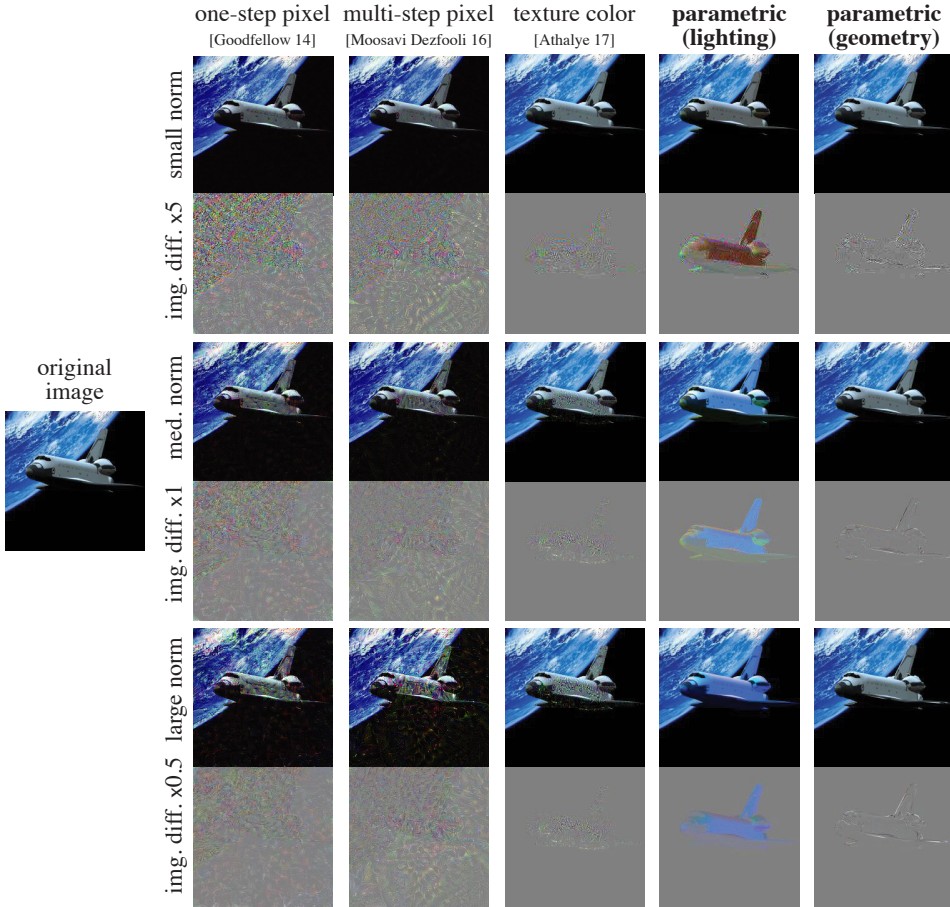

Figure 12: We compare our parametric perturbations (the first two columns) with pixel/color perturbations under the same $L^\infty$ pixel norm (small: 0.12, medium: 0.53, large: 0.82). As changing physical parameters corresponds to real-world phenomena, our parametric perturbation are more realistic.

## B  PHYSICALLY BASED RENDERING

Physically based rendering (PBR) seeks to model the flow of light, typically the assumption that there exists a collection of light sources that generate light; a camera that receives this light; and a scene that modulates the flow light between the light sources and camera (Pharr et al., 2016). What follows is a brief discussion of the general task of rendering an image from a scene description and the approximations we take in order to make our renderer efficient yet differentiable.

Computer graphics has dedicated decades of effort into developing methods and technologies to enable PBR to synthesize of photorealistic images under a large gamut of performance

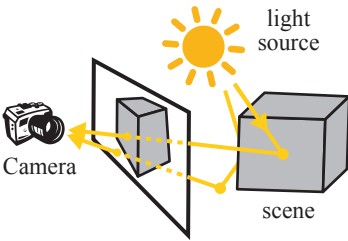

Figure 13: PBR models the physics of light that emitted from the light source, interact with the scene, then arrive a camera.

requirements. Much of this work is focused around taking approximations of the cherished Rendering equation (Kajiya, 1986), which describes the propagation of light through a point in space. If we let $u_o$ be the output radiance, $p$ be the point in space, $\omega_o$ be the output direction, $u_e$ be the emitted radiance, $u_i$ be incoming radiance, $\omega_i$ be the incoming angle, $f_r$ be the way light be reflected off the material at that given point in space we have:

$$u_o(p, \omega_o) = u_e(p, \omega_o) + \int_{S^2} f_r(p, \omega_i, \omega_o) u_i(p, \omega_i)(\omega_i \cdot \mathbf{n}) d\omega_i.$$

From now on we will ignore the emission term $u_e$ as it is not pertinent to our discussion. Furthermore, because the speed of light is substantially faster than the exposure time of our eyes, what we perceive is not the propagation of light at an instant, but the steady state solution to the rendering equation evaluated at every point in space. Explicitly computing this steady state is intractable for our applications and will mainly serve as a reference for which to place a plethora of assumptions and simplifications we will make for the sake of tractability. Many of these methods focus on ignoring light with nominal effects on the final rendered image vis a vis assumptions on the way light travels. For instance, light is usually assumed to have nominal interacts with air, which is described as the assumption that the space between objects is a vacuum, which constrains the interactions of light to the objects in a scene. Another common assumption is that light does not penetrate objects, which makes it difficult to render objects like milk and human skin[1]. This constrains the complexity of light propagation to the behavior of light bouncing off of object surfaces.

## B.1 LOCAL ILLUMINATION

It is common to see assumptions that limit number of bounces light is allowed.In our case we chose to assume that the steady state is sufficiently approximated by an extremely low number of iterations: one. This means that it seems sufficient to model the lighting of a point in space by the light sent to it directly by light sources. Working with such a strong simplification does, of course, lead to a few artifacts. For instance, light occluded by other objects is ignored so shadows disappear and auxiliary techniques are usually employed to evaluate shadows (Williams, 1978; Miller, 1994).

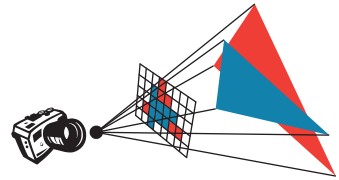

Figure 14: Rasterization converts a 3D scene into pixels.

When this assumption is coupled with a camera we approach what is used in standard *rasterization* systems such as OPENGL (Shreiner & Group, 2009), which is what we use. These systems compute the illumination of a single pixel by determining the fragment of an object visible through that pixel and only computing the light that traverses directly from the light sources, through that fragment, to that pixel. The lighting of a fragment is therefore determined by a point and the surface normal at that point, so we write the fragment's radiance as $R(p, \mathbf{n}, \omega_o) = u_o(p, \omega_o)$:

$$R(p, \mathbf{n}, \omega_o) = \int_{S^2} f_r(p, \omega_i, \omega_o) u_i(p, \omega_i)(\omega_i \cdot \mathbf{n}) d\omega_i. \tag{6}$$

## B.2 LAMBERTIAN MATERIAL

Each point on an object has a model approximating the transfer of incoming light to a given output direction $f_r$, which is usually called the material. On a single object the material parameters may vary quite a bit and the correspondence between points and material parameters is usually called the *texture map* which forms the *texture* of an object. There exists a wide gamut of material models, from mirror materials that transport light from a single input direction to a single output direction, to materials that reflect light evenly in all directions, to materials liked brushed metal that reflect differently along different angles. For the sake of document we only consider diffuse materials, also called *Lambertian* materials, where we assume

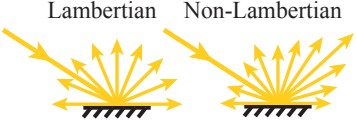

Figure 15: We consider the Lambertian material (left) where lights get reflected uniformly in every direction.

---

[1]this is why simple renderers make these sorts of objects look like plastic

that incoming light is reflected uniformly, i.e $f_r$ is a constant function with respect to angle, which we denote $f_r(p, \omega_i, \omega_o) = \rho(p)$:

$$R(p, \mathbf{n}) = \rho(p) \int_{\Omega(\mathbf{n})} u(p, \omega)(\omega \cdot \mathbf{n}) d\omega. \tag{7}$$

This function $\rho$ is usually called the *albedo*, which can be perceived as color on the surface for diffuse material, and we reduce our integration domain to the upper hemisphere $\Omega(\mathbf{n})$ in order to model light not bouncing through objects. Furthermore, since only the only $\omega$ and $u$ are the incoming ones we can now suppress the "incoming" in our notation and just use $\omega$ and $u$ respectively.

### B.3 Environment Mapping

The illumination of static, distant objects such as the ground, the sky, or mountains do not change in any noticeable fashion when objects in a scene are moved around, so $u$ can be written entirely in terms of $\omega$, $u(p, \omega) = u(\omega)$. If their illumination forms a constant it seems prudent to pre-compute or cache their contributions to the illumination of a scene. This is what is usually called *environment mapping* and they fit in the rendering equation as a representation for the total lighting of a scene, i.e the total incoming radiance $u_i$. Because the environment is distant, it is common to also assume that the position of the object receiving light from an environment map does not matter so this simplifies $u_i$ to be independent of position:

$$R(p, \mathbf{n}) = \rho(p) \int_{\Omega(\mathbf{n})} u(\omega) \, (\omega \cdot \mathbf{n}) \, d\omega. \tag{8}$$

### B.4 Spherical Harmonics

Despite all of our simplifications, the inner integral is still a fairly generic function over $S^2$. Many techniques for numerically integrating the rendering equation have emerged in the graphics community and we choose one which enables us to perform pre-computation and select a desired spectral accuracy: *spherical harmonics*. Spherical harmonics are a basis on $S^2$ so, given a spherical harmonics expansion of the integrand, the evaluation of the above integral can be reduced to a weighted product of coefficients. This particular basis is chosen because it acts as a sort of Fourier basis for functions on the sphere and so the bases are each associated with a frequency, which leads to a convenient multi-resolution structure. In fact, the rendering of diffuse objects under distant lighting can be 99% approximated by just the first few spherical harmonics bases (Ramamoorthi & Hanrahan, 2001).

We will only need to note that the spherical harmonics bases $Y_l^m$ are denoted with the subscript with $l$ as the frequency and that there are $2l + 1$ functions per frequency, denoted by superscripts $m$ between $-l$ to $l$ inclusively. For further details on them please take a glance at Appendix C.

If we approximate a function $f$ in terms of spherical harmonics coefficients $f \approx \sum_{lm} f_{l,m} Y_l^m$ the integral can be precomputed as

$$\int_{S^2} f \approx \int_{S^2} \sum_{lm} f_{l,m} Y_l^m = \sum_{lm} f_{l,m} \int_{S^2} Y_l^m, \tag{9}$$

Thus we have defined a reduced rendering equation that can be efficiently evaluated using OpenGL while maintaining differentiability with respect to lighting and vertices. In the following appendix we will derive the derivatives necessary to implement our system.

## C Differentiable Renderer

Rendering computes an image of a 3D shape given lighting conditions and the prescribed material properties on the surface of the shape. Our differentiable renderer assumes Lambertian reflectance, distant light sources, local illumination, and piece-wise constant textures. We will discuss how to explicitly compute the derivatives used in the main body of this text. Here we give a detailed discussion about spherical harmonics and their advantages.

## C.1 SPHERICAL HARMONICS

Spherical harmonics are usually defined in terms of the Legendre polynomials, which are a class of orthogonal polynomials defined by the recurrence relation

$$P_0 = 1 \tag{10}$$
$$P_1 = x \tag{11}$$
$$(l+1)P_{l+1}(x) = (2l+1)xP_l(x) - lP_{l-1}(x). \tag{12}$$

The *associated* Legendre polynomials are a generalization of the Legendre polynomials and can be fully defined by the relations

$$P_l^0 = P_l \tag{13}$$
$$(l-m+1)P_{l+1}^m(x) = (2l+1)xP_l^m(x) - (l+m)P_{l-1}^m(x) \tag{14}$$
$$2mxP_l^m(x) = -\sqrt{1-x^2}\left[P_l^{m+1}(x) + (l+m)(l-m+1)P_l^{m-1}(x)\right]. \tag{15}$$

Using the associated Legendre polynomials $P_l^m$ we can define the spherical harmonics basis as

$$Y_l^m(\theta, \phi) = K_l^m \begin{cases} (-1)^m \sqrt{2} P_l^{-m}(\cos\theta)\sin(-m\phi) & m < 0 \\ (-1)^m \sqrt{2} P_l^m(\cos\theta)\cos(m\phi) & m > 0 \\ P_l^0(\cos\theta) & m = 0 \end{cases}. \tag{16}$$

$$\text{where } K_l^m = \sqrt{\frac{(2l+1)(l-|m|)!}{4\pi(l+|m|)!}}. \tag{17}$$

We will use the fact that the associated Legendre polynomials correspond to the spherical harmonics bases that are rotationally symmetric along the $z$ axis ($m = 0$).

In order to incorporate spherical harmonics into Equation 8, we change the integral domain from the upper hemisphere $\Omega(\mathbf{n})$ back to $S^2$ via a max operation

$$R(p, \mathbf{n}) = \rho(p) \int_{\Omega(\mathbf{n})} u(\omega)(\omega \cdot \mathbf{n})d\omega \tag{18}$$

$$= \rho(p) \int_{S^2} u(\omega)\max(\omega \cdot \mathbf{n}, 0)d\omega. \tag{19}$$

We see that the integral is comprised of two components: a lighting component $u(\omega)$ and a component that depends on the normal $\max(\omega \cdot \mathbf{n}, 0)$. The strategy is to pre-compute the two components by projecting onto spherical harmonics, and evaluating the integral via a dot product at runtime, as we will now derive.

## C.2 LIGHTING IN SPHERICAL HARMONICS

Approximating the lighting component $u(\omega)$ in Equation 19 using spherical harmonics $Y_l^m$ up to band $n$ can be written as

$$u(\omega) \approx \sum_{l=0}^{n} \sum_{m=-l}^{l} U_{l,m} Y_l^m(\omega),$$

where $U_{l,m} \in \mathbb{R}$ are coefficients. By using the orthogonality of spherical harmonics we can use evaluate these coefficients as an integral between $u(\omega)$ and $Y_l^m(\omega)$

$$U_{l,m} = \langle u, Y_l^m \rangle_{S^2} = \int_{S^2} u(\omega) Y_l^m(\omega)d\omega,$$

which can be evaluated via quadrature.

## C.3 CLAMPED COSINE IN SPHERICAL HARMONICS

So far, we have projected the lighting term $u(\omega)$ onto the spherical harmonics basis. To complete evaluating Equation 19 we also need to approximate the second component $\max(\omega \cdot \mathbf{n}, 0)$ in spherical

harmonics. This is the so-called the *clamped cosine* function.

$$g(\omega, \mathbf{n}) = \max(\omega \cdot \mathbf{n}, 0) = \sum_{l=0}^{n} \sum_{m=-l}^{l} G_{l,m}(\mathbf{n}) Y_l^m(\omega),$$

where $G_{l,m}(\mathbf{n}) \in \mathbb{R}$ can be computed by projecting $g(\omega, \mathbf{n})$ onto $Y_l^m(\omega)$

$$G_{l,m}(\mathbf{n}) = \int_{\mathcal{S}^2} \max(\omega \cdot \mathbf{n}, 0) Y_l^m(\omega) d\omega.$$

Unfortunately, this formulation turns out to be tricky to compute. Instead, the common practice is to analytically compute the coefficients for unit $z$ direction $\tilde{G}_{l,m} = G_{l,m}(\mathbf{n}_z) = G_{l,m}([0, 0, 1]^\mathsf{T})$ and evaluate the coefficients for different normals $G_{l,m}(\mathbf{n})$ by rotating $\tilde{G}_{l,m}$. This rotation, $\tilde{G}_{l,m}$, can be computed analytically:

$$\begin{aligned}
\tilde{G}_{l,m} &= \int_{\mathcal{S}^2} \max(\omega \cdot \mathbf{n}_z, 0) Y_l^m(\omega) d\omega \\
&= \int_0^{2\pi} \int_0^{\pi} \max([\sin\theta\cos\phi, \sin\theta\sin\phi, \cos\theta][0, 0, 1]^\mathsf{T}, 0) Y_l^m(\theta, \phi) \sin\theta d\theta d\phi \\
&= \int_0^{2\pi} \int_0^{\pi} \max(\cos\theta, 0) Y_l^m(\theta, \phi) \sin\theta d\theta d\phi \\
&= \int_0^{2\pi} \int_0^{\pi/2} \cos\theta \, Y_l^m(\theta, \phi) \sin\theta d\theta d\phi.
\end{aligned} \tag{20}$$

In fact, because $\max(\omega \cdot \mathbf{n}_z, 0)$ is rotationally symmetric around the $z$-axis, its projection onto $Y_l^m(\omega)$ will have many zeros except the rotationally symmetric spherical harmonics $Y_l^0$. In other words, $\tilde{G}_{l,m}$ is non-zero only when $m = 0$. So we can simplify Equation 20 to

$$\tilde{G}_l = \tilde{G}_{l,0} = 2\pi \int_0^{\pi/2} \cos\theta \, Y_l^0(\theta) \sin\theta d\theta.$$

The evaluation of this integral can be found in Appendix A in (Basri & Jacobs, 2003). We provide this here as well:

$$\tilde{G}_l = \begin{cases}
\frac{\sqrt{\pi}}{2} & l = 0 \\
\sqrt{\frac{\pi}{3}} & l = 1 \\
(-1)^{\frac{l}{2}+1} \frac{(l-2)! \sqrt{(2l+1)\pi}}{2^l (\frac{l}{2}-1)! (\frac{l}{2}+1)!} & l \geq 2, \text{even} \\
0 & l \geq 2, \text{odd}
\end{cases}.$$

The spherical harmonics coefficients $G_{l,m}(\mathbf{n})$ of the clamped cosine function $g(\omega, \mathbf{n})$ can be computed by rotating $\tilde{G}_l$ (Sloan et al., 2005) using this formula

$$G_{l,m}(\mathbf{n}) = \sqrt{\frac{4\pi}{2l+1}} \tilde{G}_l \, Y_l^m(\mathbf{n}). \tag{21}$$

So far we have projected the two terms in Equation 19 into the spherical harmonics basis. Orthogonality of spherical harmonics makes the evaluation of this integral straightforward:

$$\begin{aligned}
\int_{\mathcal{S}^2} u(\omega) \max(\omega \cdot \mathbf{n}, 0) d\omega &= \int_{\mathcal{S}^2} \left[ \sum_{l,m} U_{l,m} Y_l^m(\omega) \right] \left[ \sum_{j,k} G_{j,k}(\mathbf{n}) Y_j^k(\omega) \right] d\omega \\
&= \sum_{j,k,l,m} U_{l,m} G_{j,k}(\mathbf{n}) \delta_j^l \delta_k^m \tag{22} \\
&= \sum_{l,m} U_{l,m} G_{l,m}(\mathbf{n}). \tag{23}
\end{aligned}$$

This, in conjunction with Equation 21 allows us to derive the rendering equation using spherical harmonics lighting for Lambertian objects:

$$R(p, \mathbf{n}) = \rho(p) \sum_{l=0}^{n} \sum_{m=-l}^{l} U_{l,m} \sqrt{\frac{4\pi}{2l+1}} \tilde{G}_l \, Y_l^m(\mathbf{n}). \tag{24}$$

So far we have only considered the shading of a specific point $p$ with surface normal $\mathbf{n}$. If we consider the rendered image $I$ given a shape $V$, lighting $U$, and camera parameters $\eta$, the image $I$ is the evaluation of the rendering equation $R$ of each point in $V$ visible through each pixel in the image. This pixel to point mapping is determined by $\eta$. Therefore, we can write $I$ as

$$I(V, U, \eta) = \rho(V, \eta) \underbrace{\sum_{l=0}^{n} \sum_{m=-l}^{l} U_{l,m} \sqrt{\frac{4\pi}{2l+1}} \tilde{G}_l \, Y_l^m(N(V))}_{F(V,U)}, \tag{25}$$

where $N(V)$ is the surface normal. We exploit the notation and use $\rho(V, \eta)$ to represent the texture of $V$ mapped to the image space through $\eta$.

### C.4 Lighting and Texture Derivatives

For our applications we must differentiate Equation 25 with respect to lighting and material parameters. The derivative with respect to the lighting coefficients $U$ can be obtained by

$$\frac{\partial I}{\partial U} = \frac{\partial \rho}{\partial U} F + \rho \frac{\partial F}{\partial U} \tag{26}$$

$$= 0 + \rho \sum_{l=0}^{n} \sum_{m=-l}^{l} \frac{\partial F}{\partial U_{l,m}}. \tag{27}$$

This is the Jacobian matrix that maps from spherical harmonics coefficients to pixels. The term $\partial F / \partial U_{l,m}$ can then be computed as

$$\frac{\partial F}{\partial U_{l,m}} = \sqrt{\frac{4\pi}{2l+1}} \tilde{G}_l \, Y_l^m(N(V)). \tag{28}$$

The derivative with respect to texture is defined by

$$\frac{\partial I}{\partial \rho} = \sum_{l=0}^{n} \sum_{m=-l}^{l} U_{l,m} \sqrt{\frac{4\pi}{2l+1}} \tilde{G}_l \, Y_l^m(N(V)). \tag{29}$$

Note that we assume texture variations are piece-wise constant with respect to our triangle mesh discretization.

## D Differentiating Skylight Parameters

To model possible outdoor daylight conditions, we use the analytical Preetham skylight model (Preetham et al., 1999). This model is calibrated by atmospheric data and parameterized by two intuitive parameters: turbidity $\tau$, which describes the cloudiness of the atmosphere, and two polar angles $\theta_s \in [0, \pi/2], \phi_s \in [0, 2\pi]$, which are encode the direction of the sun. Note that $\theta_s, \phi_s$ are not the polar angles $\theta, \phi$ for representing incoming light direction $\omega$ in $u(\omega)$. The spherical harmonics representation of the Preetham skylight is presented in (Habel et al., 2008) as

$$u(\omega) = \sum_{l=0}^{6} \sum_{m=-l}^{l} U_{l,m}(\theta_s, \phi_s, \tau) Y_l^m(\omega).$$

This is derived by first performing a non-linear least squares fit to write $U_{l,m}$ as a polynomial of $\theta_s$ and $\tau$ which lets them solve for $\tilde{U}_{l,m}(\theta_s, \tau) = U_{l,m}(\theta_s, 0, \tau)$

$$\tilde{U}_{l,m}(\theta_s, \tau) = \sum_{i=0}^{13} \sum_{j=0}^{7} (p_{l,m})_{i,j} \theta_s^i \tau^j,$$

where $(p_{l,m})_{i,j}$ are scalar coefficients, then $U_{l,m}(\theta_s, \phi_s, \tau)$ can be computed by applying a spherical harmonics rotation with $\phi_s$ using

$$U_{l,m}(\theta_s, \phi_s, \tau) = \tilde{U}_{l,m}(\theta_s, \tau) \cos(m\phi_s) + \tilde{U}_{l,-m}(\theta_s, \tau) \sin(m\phi_s).$$

We refer the reader to (Preetham et al., 1999) for more detail. For the purposes of this article we just need the above form to compute the derivatives.

### D.1 Derivatives

The derivatives of the lighting with respect to the skylight parameters $(\theta_s, \phi_s, \tau)$ are

$$\frac{\partial U_{l,m}(\theta_s, \phi_s, \tau)}{\partial \phi_s} = -m\tilde{U}_{l,m}(\theta_s, \tau) \sin(m\phi_s) + m\tilde{U}_{l,-m}(\theta_s, \tau) \cos(m\phi_s) \tag{30}$$

$$\frac{\partial U_{l,m}(\theta_s, \phi_s, \tau)}{\partial \theta_s} = \frac{\partial \tilde{U}_{l,m}(\theta_s, \tau) \cos(m\phi_s) + \tilde{U}_{l,-m}(\theta_s, \tau) \sin(m\phi_s)}{\partial \theta_s} \tag{31}$$

$$= \sum_{ij} i\theta_s^{i-1}\tau^j (p_{l,m})_{i,j} \cos(m\phi_s) + \sum_{ij} i\theta_s^{i-1}(p_{l,-m})_{i,j} \sin(m\phi_s) \tag{32}$$

$$\frac{\partial U_{l,m}(\theta_s, \phi_s, \tau)}{\partial \tau} = \sum_{ij} j\theta_s^i \tau^{j-1}(p_{l,m})_{i,j} \cos(m\phi_s) + \sum_{ij} j\theta_s^i \tau^{j-1}(p_{l,-m})_{i,j} \sin(m\phi_s) \tag{33}$$

## E  Derivatives of Surface Normals

Taking the derivative of the rendered image $I$ with respect to surface normals $N$ is an essential task for computing the derivative of $I$ with respect to the geometry $V$. Specifically, the derivative of the rendering equation Equation 25 with respect to $V$ is

$$\frac{\partial I}{\partial V} = \frac{\partial \rho}{\partial V}F + \rho\frac{\partial F}{\partial V} \tag{34}$$

$$= \frac{\partial \rho}{\partial V}F + \rho\frac{\partial F}{\partial N}\frac{\partial N}{\partial V} \tag{35}$$

We assume the texture variations are piece-wise constant with respect to our triangle mesh discretization and omit the first term $\partial \rho / \partial V$ as the magnitude is zero. Computing $\partial N / \partial V$ is provided in Section 3.2. Computing $\partial F / \partial N_i$ on face $i$ is

$$\frac{\partial F}{\partial N_i} = \sum_{l=0}^{n} \sum_{m=-l}^{l} U_{l,m}\sqrt{\frac{4\pi}{2l+1}}\tilde{G}_l\frac{\partial Y_l^m}{\partial N_i}, \tag{36}$$

where the $\partial Y_l^m / \partial N_i$ is the derivative of the spherical harmonics with respect to the face normal $N_i$.

To begin this derivation recall the relationship between a unit normal vector $\mathbf{n} = (n_x, n_y, n_z)$ and its corresponding polar angles $\theta, \phi$

$$\theta = \cos^{-1}\left(\frac{n_z}{\sqrt{n_x^2 + n_y^2 + n_z^2}}\right) \qquad \phi = \tan^{-1}\left(\frac{n_y}{n_x}\right),$$

we can compute the derivative of spherical harmonics with respect to the normal vector through

$$
\frac{\partial Y_l^m(\theta, \phi)}{\partial \mathbf{n}}
$$

$$
= K_l^m \begin{cases}
(-1)^m \sqrt{2} \left[ \dfrac{\partial P_l^{-m}(\cos\theta)}{\partial\theta} \dfrac{\partial\theta}{\partial\mathbf{n}} \sin(-m\phi) + P_l^{-m}(\cos\theta) \dfrac{\partial\sin(-m\phi)}{\partial\phi} \dfrac{\partial\phi}{\partial\mathbf{n}} \right] & m < 0 \\[3mm]
(-1)^m \sqrt{2} \left[ \dfrac{\partial P_l^{m}(\cos\theta)}{\partial\theta} \dfrac{\partial\theta}{\partial\mathbf{n}} \cos(m\phi) + P_l^{m}(\cos\theta) \dfrac{\partial\cos(m\phi)}{\partial\phi} \dfrac{\partial\phi}{\partial\mathbf{n}} \right] & m > 0 \\[3mm]
\dfrac{\partial P_l^{0}(\cos\theta)}{\partial\theta} \dfrac{\partial\theta}{\partial\mathbf{n}} & m = 0
\end{cases}
$$

$$
= K_l^m \begin{cases}
(-1)^m \sqrt{2} \left[ \dfrac{\partial P_l^{-m}(\cos\theta)}{\partial\theta} \dfrac{\partial\theta}{\partial\mathbf{n}} \sin(-m\phi) - m P_l^{-m}(\cos\theta)\cos(-m\phi) \dfrac{\partial\phi}{\partial\mathbf{n}} \right] & m < 0 \\[3mm]
(-1)^m \sqrt{2} \left[ \dfrac{\partial P_l^{m}(\cos\theta)}{\partial\theta} \dfrac{\partial\theta}{\partial\mathbf{n}} \cos(m\phi) - m P_l^{m}(\cos\theta)\sin(m\phi) \dfrac{\partial\phi}{\partial\mathbf{n}} \right] & m > 0 \\[3mm]
\dfrac{\partial P_l^{0}(\cos\theta)}{\partial\theta} \dfrac{\partial\theta}{\partial\mathbf{n}} & m = 0
\end{cases}
\tag{37}
$$

Note that the derivative of the associated Legendre polynomials $P_l^m(\cos\theta)$ can be computed by applying the recurrence formula Dunster (2010)

$$
\begin{aligned}
\frac{\partial P_l^m(\cos\theta)}{\partial\theta} &= \frac{-\cos\theta(l+1)P_l^m(\cos\theta) + (l-m+1)P_{l+1}^m(\cos\theta)}{\cos^2\theta - 1} \times (-\sin\theta) \\
&= \frac{-\cos\theta(l+1)P_l^m(\cos\theta) + (l-m+1)P_{l+1}^m(\cos\theta)}{\sin\theta}.
\end{aligned}
\tag{38}
$$

Thus the derivatives of polar angles $(\theta, \phi)$ with respect to surface normals $\mathbf{n} = [n_x, n_y, n_z]$ are

$$
\frac{\partial\theta}{\partial\mathbf{n}} = \left[ \frac{\partial\theta}{\partial n_x}, \frac{\partial\theta}{\partial n_y}, \frac{\partial\theta}{\partial n_z} \right] = \frac{\left[ n_x n_z,\ n_y n_z,\ -(n_x^2 + n_y^2) \right]}{(n_x^2 + n_y^2 + n_z^2)\sqrt{n_x^2 + n_y^2}},
\tag{39}
$$

$$
\frac{\partial\phi}{\partial\mathbf{n}} = \left[ \frac{\partial\phi}{\partial n_x}, \frac{\partial\phi}{\partial n_y}, \frac{\partial\phi}{\partial n_z} \right] = \left[ \frac{-n_y}{n_x^2 + n_y^2},\ \frac{n_x}{n_x^2 + n_y^2},\ 0 \right].
\tag{40}
$$

In summary, the results of Equation 37, Equation 38, Equation 39, and Equation 40 tell us how to compute $\partial Y_l^m / \partial N_i$. Then the derivative of the pixel $j$ with respect to vertex $p$ which belongs to face $i$ can be computed as

$$
\begin{aligned}
\frac{\partial I_j}{\partial V_p} &\approx \rho_j \frac{\partial F}{\partial N_i} \frac{\partial N_i}{\partial V_p} \\
&= \rho_j \sum_{l=0}^{n} \sum_{m=-l}^{l} U_{l,m} \sqrt{\frac{4\pi}{2l+1}} \tilde{G}_l \frac{\partial Y_l^m(\theta, \phi)}{\partial N_i} \frac{\partial N_i}{\partial V_p}.
\end{aligned}
\tag{41}
$$

## F  ADVERSARIAL TRAINING IMPLEMENTATION DETAIL

Our adversarial training is based on the basic idea of injecting adversarial examples into the training set at each step and continuously updating the adversaries according to the current model parameters (Goodfellow et al., 2015; Kurakin et al., 2017). Our experiments inject 100 adversarial lighting examples to the CIFAR-100 data ($\approx 0.17\%$ of the training set) and keep updating these adversaries at each epoch.

We compute the adversarial lighting examples using the orange models collected from cgtrader.com and turbosquid.com. We uses five gray-scale background colors with intensities 0.0, 0.25, 0.5, 0.75, 1.0 to mimic images in the CIFAR-100 which contains many pure color backgrounds. Our orthographic cameras are placed at polar angle $\theta = \pi/3$ with 10 uniformly sampled azimuthal angles ranging from $\phi = 0$ to $2\pi$. Our initial spherical harmonics lighting is the same as

CIFAR-100    random light   adv. light (early epochs)  adv. light (late epochs)

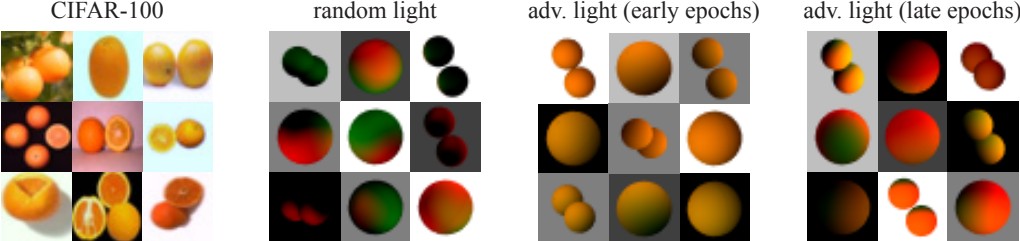

Figure 16: This figure visualizes the images of oranges from CIFAR-100, random lighting, and adversarial lighting. In early training stage, small changes in lighting are sufficient to construct adversarial examples. In late training stage, we require more dramatic changes as the model is becoming robust to differ lightings.

other experiments, using the real-world lighting data provided in (Ramamoorthi & Hanrahan, 2001). Our stepsize for computing adversaries is 0.05 along the direction of lighting gradients. We run our adversarial lighting iterations until fooling the network or reaching the maximum 30 iterations to avoid too extreme lighting conditions, such as turning the lights off.

Our random lighting examples are constructed at each epoch by randomly perturb the lighting coefficients ranging from -0.5 to 0.5.

When training the 16-layers WideResNet (Zagoruyko & Komodakis, 2016) with wide-factor 4, we use batch size 128, learning rate 0.125, dropout rate 0.3, and the standard cross entropy loss. We implement the training using PyTorch (Paszke et al., 2017), with the SGD optimizer and set the Nesterov momentum 0.9, weight decay 5e-4. We train the model for 150 epochs and use the one with best accuracy on the validation set. Figure 16 shows examples of our adversarial lights at different training stages. In the early stages, the model is not robust to different lighting conditions, thus small lighting perturbations are sufficient to fool the model. In the late stages, the network becomes more robust to different lightings. Thus it requires dramatic changes to fool a model or even fail to fool the model within 30 iterations.

## G  EVALUATE RENDERING QUALITY

We evaluated our rendering quality by whether our rendered images are recognizable by models trained on real photographs. Although large 3D shape datasets, such as ShapeNet (Chang et al., 2015), are available, they do not have have geometries or textures at the resolutions necessary to create realistic renderings. We collected 75 high-quality textured 3D shapes from cgtrader.com and turbosquid.com to evaluate our rendering quality. We augmented the shapes by changing the field of view, backgrounds, and viewing directions, then keep the configurations that were correctly classified by a pre-trained ResNet-101 on ImageNet. Specifically, we place the centroid, calculated as the weighted average of the mesh vertices where the weights are the vertex areas, at the origin and normalize shapes to range -1 to 1; the field of view is chosen to be 2 and 3 in the same unit with the normalized shape; background images include plain colors

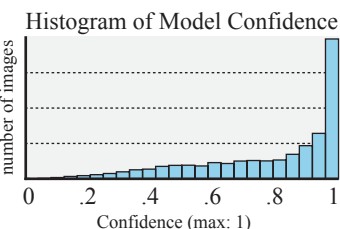

Figure 17: Prediction confidence on rendered images, showing our rendering quality is faithful enough to be confidently recognized by ImageNet models.

and real photos, which have small influence on model predictions; viewing directions are chosen to be 60 degree zenith and uniformly sampled 16 views from 0 to $2\pi$ azimuthal angle. In Figure 17, we show that the histogram of model confidence on the correct labels over 10,000 correctly classified rendered images from our differentiable renderer. The confidence is computed using softmax function and the results show that our rendering quality is faithful enough to be recognized by models trained on natural images.

