# OpenReview forum: "Beyond Pixel Norm-Balls: Parametric Adversaries using an Analytically Differentiable Renderer"
_ICLR.cc/2019/Conference_

### Official Review · AnonReviewer3 · 2018-10-31
**Interesting idea but lacks comparison with state of the art**

**Rating:** 6
**Confidence:** 3

**Review:**

Summary:
This work presents a method to generate adversary examples capable of fooling a neural network classifier. Szegedy et al. (2013) were the first to expose the weakness of neural networks against adversarial attacks, by adding a human-imperceptible noise to images to induce misclassification. Since then, several works tackled this problem by modifying the image directly in the pixel space: the norm-balls convention. The authors argue that this leads to non-realistic attacks and that a network would not benefit from training with these adversarial images when performing in the real world. Their solution and contributions are parametric norm-balls: unlike state-of-the-art methods, they perform perturbations in the image formation space, namely the geometry and the lighting, which are indeed perturbations that could happen in real life. For that, they defined a differentiable renderer by making some assumptions to simplify its expression compared to solving a light transport equation. The main simplifications are the direct illumination to gain computation efficiency and the distant illumination and diffuse material assumptions to represent lighting in terms of spherical harmonics as in Ramamoorthi et al. (2001), which require only 9 parameters to approximate lighting. This allows them to analytically derivate their loss function according to the geometry and lighting and therefore generate their adversary examples via gradient descent. They show that their adversary images generalize to other classifiers than the one used (ResNet). They then show that injecting these images into the training set increase the robustness of WideResNet against real attacks. These real attack images were taken by the authors in a laboratory with varying illumination.

Strength:
- The proposed perturbations in the image formation space simulate the real life scenario attacks.
- The presented results show that the generated adversary images do fool the classifier (used to compute the loss) but also new classifiers (different than the one used to compute the loss). As a consequence the generated adversary images increase the robustness of the considered classifier.
- Flexibility in their cost function allows for diverse types of attacks: the same modified geometry can fool a classifier in several views, either into detecting the same object or detecting different false objects under different views.

Major comments:
- Method can only compute synthetic adversary examples, unlike state-of-the-art.
- The main contribution claimed by the author is that their perturbations are realistic and that it would help better increase the robustness of classifiers against real attacks. However, they do not give any comparison to the state-of-the-art methods as is expected.

Minor comments:
- Even if the paper is well written, they are still some typos.

---

> ### Author Response · Authors · 2018-11-15
> **Authors' Reply**
>
>
> # Comparisons with state-of-the-art
> We include direct comparisons to state-of-the-art differentiable renderers in Section 2 and Section 3.1. These clearly demonstrate our superiority with respect to speed and memory.
>
> Conducting direct comparisons to state-of-the-art adversarial attacks is less well-posed. In the revision, we have expanded our feature comparison with a new table in Section 2. See further discussion in the Revision Summary post above.

---

> > ### Comment · AnonReviewer3 · 2018-11-26
> > **Reviewer 3 comments**
> >
> > Thank you for the rebuttal. I think my initial rating is still relevant even after the revisions of the authors.

---

### Official Review · AnonReviewer1 · 2018-11-03
**Good paper, but please address questions**

**Rating:** 7
**Confidence:** 4

**Review:**

The paper demonstrates a method for constructing adversarial examples by modifications or perturbations to physical parameters in the scene itself---specifically scene lighting and object geometry---such that images taken of that scene are able to fool a classifier. It achieves this through a novel differentiable rendering engine, which allows the proposed method to back-propagate gradients to the desired physical parameters. Also interesting in the paper is the use of spherical harmonics, which restrict the algorithm to plausible lighting. The method is computationally efficient and appears to work well, generating plausible scenes that fool a classifier when imaged from different viewpoints.

Overall, I have a positive view of the paper. However, there are certain issues below that the authors should address in the rebuttal for me to remain with my score of accept (especially the first one):


- The paper has no discussion of or comparisons to the work of Athalye and Sutskever, 2017 and Zeng et al., 2017, except for a brief mention in Sec 2 that these methods also use differentiable renderers for adversarial attacks. These works address the same problem as this paper---computing physically plausible adversarial attacks---and by very similar means---back-propagation through a rendering engine. Therefore it is critical that the paper clarifies its novelty over these methods, and if appropriate, include comparisons.

- While the goal of finding physically plausible adversarial examples is indeed important, I disagree with the claim that image-level attacks are "primarily tools of basic research, and not models of real-world security scenarios". In many applications, an attacker may have access to and be able to modify images after they've been captured and prior to sending them through a classifier (e.g., those attempting to detect transmission of spam or sensitive images). I believe the paper can make its case about the importance of physical adversarial perturbations without dismissing image-level perturbations as entirely impractical.

- The Athalye 18 reference noted in Fig 1 is missing (the references section includes the reference to Athalye and Sutskever '17).

===Post-rebuttal

Thanks for addressing my questions. With the new comparisons and discussions wrt the most relevant methods, I believe the contributions of the paper are clearer. I'm revising my score from 6 to 7.

---

> ### Author Response · Authors · 2018-11-15
> **Authors' Reply**
>
>
> # Comparisons
> See the Revision Summary post regarding a new comparison table.
>
> # Image-level perturbations
> We have toned down our statements in the introduction.

---

### Official Review · AnonReviewer2 · 2018-11-04
**The paper describes the use of differentiable physics based rendering schemes to generate adversarial perturbations that are constrained by physics of image formation. The paper demonstrates how data augmentation using the scheme can improve robustness of classifiers in a limited experimental setting.**

**Rating:** 7
**Confidence:** 4

**Review:**

Quality of the paper:  The paper is quite clear on the background literature on adversarial examples, physics based rendering, and the core idea of generating adversarial perturbations as a function of illumination and geometric changes.
Originality and Significance: The idea of using differential renderers to produce physically consistent adversarial perturbations is novel.
References: The references in the paper given its scope is fine.  It is recommended to  explore references to other recent papers that use simulation for performance enhancement in the context of transfer learning, performance characterization (e.g. veerasavarappu et al in arxiv, WACV, CVPR (2015 - 17))

Pros:  Good paper , illustrates the utility of differentiable rendering and simulations to generate adversarial examples and to use them for improving robustness.
Cons: The experimental section needs to be extended and the results are limited to simulations on CIFAR-100 and evaluation on lab experimental data.  Inclusion of images showing CIFAR-100 images augmented with random lighting, adversarial lighting would have been good. The details of the image generation process for that experiment is vague and not reproducible.

---

> ### Author Response · Authors · 2018-11-15
> **Authors' Reply**
>
>
> # Simulation for performance enhancement
> We thank the reviewer for pointing out related papers on this topic. They led us to many papers that demonstrate that models trained on synthetic data can outperform those trained on real data alone for real-world tasks. These references further strengthen our case for moving beyond the pixel-ball norms (Section 2, 5, and 6; highlighted in green).
>
> # Adversarial training
> Our paper focuses on how to create adversarial attacks beyond the pixel norm-ball using physical parameters via a novel differentiable renderer. In the revision, we have improved the description of our preliminary application of this insight to adversarial training (a replicable description is now provided in Appendix F). A more exhaustive study of adversarial training is left as future work (additional discussion in Section 6).

---

### Author Response · Authors · 2018-11-15
**Revision Summary**

Thank you for your helpful comments and enthusiasm. In the revised document, we highlight all the changes in the green text. Our major changes are:
# Add references that use simulation to enhance network performance on real-world tasks (Section 2)
# Add detail of the adversarial training (Appendix F)
# Add future extension for the adversarial training (Section 6)
# Add a table comparison with previous non-image based adversarial attacks (Section 2)
# Tone down the argument on image-based adversarial attacks (Section 1)
# Typographical and reference issues

# Comparison Feature Table (R1, R3)
We have included a new feature comparison table in Section 2 highlighted in green. This table shows that while [Athalye 2017] generates adversarial colors on the surface geometry, that method cannot compute adversarial examples by perturbing the physical parameters we are focusing on (lighting and geometry). Therefore, our methods are complementary.
Meanwhile, [Zeng 2017] requires a non-trivial training phase to learn a proxy renderer. This training requires a substantial amount of data. Further, this data should be representative of scenes that will be witnessed at runtime, otherwise training-bias will occur. Even assuming high-quality training, the method of [Zeng 2017]  still takes orders of magnitude longer to compute adversarial examples (12 minutes reported in [Zeng 2017] versus a few seconds using our method).

---

> ### Author Response · Authors · 2018-11-25
> **Thanks for the replies**
>
> We changed the color of the updated text from green back to black as tomorrow is the end of the revision period. Thank you for all the replies.

---

### Public Comment · (anonymous) · 2018-11-26
**Concerns with "believability" and other claims**

The adversarial examples created here with the differentiable renderer are certainly cool. However I have some concerns with the claimed contributions.

First, a key claimed contribution is that of believability. To test this, the authors set a given \ell_\infty \epsilon threat model, and generate adversarial examples with each method. This is a flawed experiment: by fixing a threat model, the compared against methods will use the entire allowed perturbation bound to create adversarial examples. All methods are minor variations of Projected Gradient Descent, which will, by the nature of the underlying algorithm, use the whole allowed perturbation budget (i.e. will produce perturbations that extend to the edges of the allowed \ell_\infty box). Therefore the experiment in Figure 1 shows nothing about the "believability" of perturbations produced with the various methods (the spaceship here could likely be misclassified with an imperceptible \ell_\infty norm perturbation generated with PGD).

The authors also claim that their method extends to create physical world adversarial examples, but only show this with adversarial lighting (not color, geometry, or any of the other parameters listed in Table 1) on a single example (oranges) in front of a single, uniformly black, backdrop, at a single angle.

Also, the citation for Athalye et al (listed as Athalye and Sutskever) is wrong; it should be:

@misc{athalye2017synthesizing,
    title={Synthesizing Robust Adversarial Examples},
    author={Anish Athalye and Logan Engstrom and Andrew Ilyas and Kevin Kwok},
    year={2017},
    eprint={1707.07397},
    archivePrefix={arXiv},
    primaryClass={cs.CV}
}

---

> ### Author Response · Authors · 2018-11-28
> **Re: Concerns with "believability" and other claims**
>
> Indeed, current pixel-based attacks can fool classifiers with imperceivable perturbations. The magnitude of a perturbation is not the only factor that determines how realistic or plausible it is to occur in the real world. Figure 1 demonstrates, reductio ad absurdum, that very large pixel perturbations can be realistic if the perturbation is conducted in the physical parameter space (e.g., lighting). We have provided visualization of small perturbations in Figure 12. Specifically, Figure 12 shows perturbations with the same \ell_\infty norm across columns and magnifies the perturbations at each row differently for visualization purposes. However, the structure of imperceivable perturbation may still not correspond to any real-world scenario.
>
> Our claimed contribution is to construct adversarial examples through perturbing physical parameters of the image formation model. We leave physical world geometry attacks to future work as it involves a non-trivial computational fabrication engineering aspect.
>
> Thanks, we've corrected the reference

---

### Meta-Review · Area_Chair1 · 2018-12-15
**An interesting contribution, although some concerns regarding the claims**

**Confidence:** 4
**Recommendation:** Accept (Poster)

**Metareview:**

The paper describes the use of differentiable physics based rendering schemes to generate adversarial perturbations that are constrained by physics of image formation.

The paper puts forth a fairly novel approach to tackle an interesting question. However, some of the claims made regarding the "believability" of the adversarial examples produced by existing techniques are not fully supported. Also, the adversarial examples produced by the proposed techniques are not fully "physical" at least compared to how "physical" adversarial examples presented in some of the prior work were.

Overall though this paper constitutes a valuable contribution.